# Generating Memorable Mnemonic Encodings of Numbers

## Abstract

The major system is a mnemonic system that can be used to memorize sequences of numbers. In this work, we present a method to automatically generate sentences that encode a given number. We propose several encoding models and compare the most promising ones in a password memorability study. The results of the study show that a model combining part-of-speech sentence templates with an $n$-gram language model produces the most memorable password representations.

## 1 Introduction

The major system is a mnemonic device used to help memorize numbers. The system works by mapping each digit of a number to a consonant phoneme and allowing for arbitrary insertion of vowel phonemes to produce words (Fauvel-Gouraud, 1845). For instance, the digit 1 maps to $<T>$, and the digit 2 maps to $<N>$. The number 121 can then be encoded as the word *tent* by replacing both 1s with $<T>$s, replacing the 2 with $<N>$, and inserting an $<e>$. The full major system mapping is shown in Table 1.

The difficulty of generating a memorable sequence of words that encodes a number with the major system stems from the constraint that the sequence of words must encode exactly the given digits. While there are many sequences of words that correctly encode a given number, the vast majority of these sequences are incoherent and thus difficult to remember. So, this task requires the use of a language model that balances the encoding constraints with syntactic plausibility and some notion of memorability.

We have developed a system that automatically produces a sequence of words to encode a se-

| Digit | Corresponding Phonemes |
|-------|------------------------|
| 0 | S, Z |
| 1 | T, D, TH, DH |
| 2 | N |
| 3 | M |
| 4 | R |
| 5 | L |
| 6 | CH, JH, SH, ZH |
| 7 | K, G |
| 8 | F, V |
| 9 | P, B |
| None | NG, vowels |

Table 1: The major system maps digits to Arpabet consonant phonemes.

quence of digits. Each such encoding is a sequence of sentences that balance memorability and length. We sample from a distribution of part-of-speech (POS) templates to produce a syntactically plausible sentence, then use an $n$-gram language model to fill each POS slot in the selected template to produce an encoding.

A system like ours can be used to memorize fairly short numbers, such as a numeric password, a phone number, or an account number; or to memorize arbitrarily long numbers, like digits of $\pi$. One could use our system to encode a smartphone passcode as a short sentence. Thus, our system can help improve the strength of security practices.

To test the effectiveness of our system, we conducted a study on password memorability. Participants were asked to memorize an eight-digit number representing a numeric password and a phrase produced by our system to encode the same number. After seven days, participants remembered the encoding produced by our final model better than the number itself. Participants also stated a

strong preference for our final model's encodings.

The rest of this paper is organized as follows. Section 2 describes existing systems for automatically generating encodings with the major system and puts our work in the context of related academic problems. In Section 3, we describe the different encoding models that we studied. Section 4 gives results and analysis of our models. In Section 5, we describe the password memorability study we conducted. We describe possible extensions to our work and conclude in Section 6.

## 2 Previous Work

Several tools are available online for generating naive encodings of numbers using the major system. In this section, we describe those tools and identify the shortcomings in those implementations that are addressed by our work. We also put our work into the context of previous studies on password memorability.

### 2.1 Existing Tools

There are a number of existing tools that use the major system to encode sequences of digits as sequences of words. However, all such tools we found have considerable limitations. Most notably, the majority of these tools simply return the entire set of words that can individually encode the given number.

Many mobile applications will generate encodings, but their focus appears to be on helping users learn the major system and not on generating memorable encodings automatically (Shichijo, 2014; Reindl, 2015; Vladislav, 2016; Scott, 2015a; Pfeiffer, 2013; Buder, 2012). None of these tools rank the multiple encodings they produce, and none of them produce sentences. Most of these tools only produce one- or two-word encodings, greatly limiting the length of sequences they can encode (Graaff, 2016; Rememberg, 2010; Jurkowski, 2014).

Other tools produce encodings of longer sequences of digits by breaking the sequence into chunks of a fixed length, often two digits per word, and most do not combine the single-word encodings into one sequence of words (Ströbele, 2013; Scott, 2015b; Parcel, 2016; Got2Know, 2013).

Thus, these existing approaches are ill-suited for the memorization of even moderately long sequences of digits. Since the most sophisticated of these approaches are equivalent to our baseline models, we do not empirically compare these tools to our models.

### 2.2 Related Academic Work

We are only aware of two previous corpus-based methods for generating mnemonic encodings. The first presents a method to help remember text passwords by finding a newspaper headline from the Reuters Corpus such that the first letters of each word in the headline match the letters in the password (Jeyaraman and Topkara, 2005). However, the restriction of using only newspaper headlines means that only about 40% of seven-character passwords are covered.

The second corpus-based method addresses the related problem of memorizing random strings of bits. Ghazvininejad and Knight (2015) created a method to encode random 60-bit strings as memorable sequences of words. However, their methods that create the most memorable passwords do not allow the user to mentally convert their memorized sequence of words to the original string. In contrast, our use of the major system allows users to easily convert any sequence of words into the encoded number in their head.

Although these methods encode a sequence of letters and a string of bits while our system encodes a sequence of digits, all aim to create memorable sentences as output. Based on the results of these two previous methods, our system favors unique words and sentences of moderate length. Because our system needs to encode any arbitrary sequence of digits, we use a language model to generate sentences instead of relying on a preexisting set of newspaper headlines.

Substantially more work has been done on the memorability and security of passwords. Forget and Biddle (2008) found that modifying user-created passwords to increase security had the unintended consequence of reducing memorability. Yan et al. (2004)'s work provides a possible means of dealing with that tension between security and memorability, showing that passwords based on mnemonic phrases were as easy to remember as naively created passwords and as strong as random passwords. Their positive results for mnemonic-based passwords are encouraging for our own mnemonic-based system. Our system is further informed by the work of Leiva and Sanchis-Trilles (2014), who analyzed different methods of sampling memorable sentences from corpora to use as

prompts in text entry research. They found that prompts are more memorable when they are complete phrases and have fewer words.

Our user study experiment evaluating the memorability of the phrases generated by our systems is informed by the existing work in this area. The format of our human subjects experiment is largely informed by the work of Vu et al. (2004). They examined the use of passphrases created by taking the first letter of each word in a sentence. Their user study split participants into two passphrase groups, the second of which had to include a number and a special character in the passphrase. The participants were not allowed to write the passwords down. The researchers then tested the participants' recall after five minutes and after a week. The results showed that the second group produced much more secure passwords but at the cost of memorability.

## 3 Methodology

In this section, we describe the data we used to train our models and present the six mnemonic-generating systems we considered. These models include two baseline models, three preliminary models, and the final sentence encoder model. The source code for these models will be made available online [1].

### 3.1 Data Sets

We use two data sets in our system. The first is the Brown corpus, which contains about 56,000 types and 1.2 million tokens (Francis and Kuera, 1964). We use this corpus to train our $n$-gram models. The corpus is also tagged with part-of-speech data, which we use to train our part-of-speech $n$-gram model and our sentence encoder model.

The second data set we use is the CMU Pronouncing Dictionary (Lenzo, 2014). This data set is a file of about 134,000 words, each labeled with its pronunciation in the Arpabet phoneme set. We work with the intersection of these two data sets, which contains about 34,000 words. This ensures that all the words produced by our language model can be pronounced from the CMU Pronouncing Dictionary. We pre-process both data sets to lowercase all words.

---

[1] URL withheld for blind review.

### 3.2 Baseline Models

We designed two baseline models to compare our results against. Each of these baselines satisfies the requirement that the sequences of words produced encode exactly the input digits. Both baselines are *greedy*: they generate encodings one word at a time. At each time step, they choose a word from the set of words that encode the maximum number of digits possible. They differ in how a word is chosen from that set:

**Random Encoder:** At each step, a word is selected at random.

**Unigram Encoder:** At each step, the word with the highest unigram probability is selected.

### 3.3 Preliminary Models

We also considered three models that were more sophisticated than our baseline models. Unlike the baseline models that greedily encode as many digits per word as possible, these models consider all words that can encode at least one digit.

$n$-**gram Encoder:** Words are generated one at a time. At each step, the next word is chosen using an $n$-gram language model with Stupid Backoff (Brants et al., 2007). We tested different combinations of hyperparameters and decided on default values of $n = 3$ and backoff factor $\alpha = 0.1$. An additional hyperparameter indicates if the model should select the word with the highest $n$-gram probability or sample from a weighted probability distribution based on $n$-gram probabilities, with the former option as the default.

**Part-of-Speech (POS) Encoder:** Words are generated one at a time, but a POS trigram model is used to restrict the set of possible words at each step so that the generated phrases are syntactically motivated. Each word is associated with the POS tag it most often has in the Brown corpus. To select each word in the encoding, the most likely POS tag is identified from the POS trigram model. From all words with that POS, we choose the word that has the greatest likelihood according to a word trigram model.

**Chunk Encoder:** Instead of generating encodings one word at a time, we generate one sentence at a time. Each sentence must match a

fixed phrase template: <noun phrase><verb phrase><noun phrase>. Additionally, each chunk encodes exactly three digits. This encoder breaks the given number into chunks of three digits and encodes each chunk as one or two words that can be parsed as the desired chunk type. For each chunk, the phrase that has the greatest likelihood according to a bigram language model is selected.

### 3.4 Final Model: Sentence Encoder

After examining the output of the three preliminary models, we combined the best elements of each into a final model, which we call the *Sentence Encoder*. This model aims to produce a variety of sentence structures that are both adequately long and reasonably likely to occur.

The sentence encoder trains a trigram model on the Brown corpus and stores the 100 most frequent sentence templates found in the corpus. A sentence template is a sequence of part-of-speech tags from the corpus's simple "universal" tag set. We filter these sentence templates to only contain sentences that have a verb, do not have any numbers or "other"-category words (like foreign words), and are guaranteed to produce at least 5 words.

To encode a sequence of digits, the sentence encoder first samples a sentence template based on the templates' frequencies in the training corpus. Then, for every part of speech in the template, the encoder selects the word that encodes at least the next digit with the most likely trigram score based on the previous words. If the encoder is unable to find a word that matches the necessary part of speech, it replaces the current sentence with a different, newly sampled sentence template. This process is repeated until all digits are encoded, possibly resulting in the end of the last sentence template being unused.

A few additional changes to this model greatly improve its performance:

- We allow nouns in place of pronouns, since there are more possible nouns than pronouns.

- We allow certain parts of speech - determiners, adjectives, and adverbs - to be skipped if no word is found that matches them.

- We weight the trigram score of each word based on how many digits it encodes. We

do this by multiplying the score by the number of digits the word encodes raised to some power, which is set to 10 by default. We previously found that the most likely sequences of words tend to encode only one or two digits per word, resulting in sentences that are long and thus less memorable.

- We run a post-processing pass over the output sentences. The post-processing pass iterates over each word, calculates the probabilities for all possible words that encode the exact same digits using a bigram language model with the preceding and following words, and replaces the original word with the most likely word.

With these four changes, the sentence encoder is able to encode all numbers as memorable, syntactically plausible sentences of a reasonable length.

As an example, consider the number 86101521. The sentence encoder first samples a sentence template, such as "<verb> <noun> <conj> <verb> <adv>." Then, the sentence encoder finds a verb that encodes at least the first digit, 8. The sentence encoder selects the verb "officiate," which has consonant sounds "<F>,<SH>,<T>," to represent 861. The remaining digits are 01521. The sentence encoder then selects the noun "wasteland," with consonant sounds "<S>,<T>,<L>,<N>,<D>," to represent 01521. So, 86101521 is encoded as "Officiate wasteland."

## 4 Model Output

For each model, we look at the encoding generated for two numbers. The first number is a random eight-digit number, and the second is the first fifty digits of $\pi$.

The eight-digit number demonstrates each model's ability to encode fairly short sequences of digits, such as a numeric password, a phone number, or an account number. Table 2 shows how each model encoded the number 86101521.

The first fifty digits of $\pi$ demonstrates each model's ability to encode an arbitrarily long sequence of digits. Table 3 shows how each model encoded the first fifty digits of $\pi$: 31415926535897932384626433832795028841971 693993751.

| Encoder | Phrase |
|---|---|
| **Random** | Vouching wits widely and |
| **Unigram** | Fish this tell and |
| $n$-**gram** | Of which the house to all. And |
| **POS** | Wife age at sea with law in the |
| **Chunk** | Half shut settle night. |
| **Sentence** | Officiate wasteland. |

Table 2: Encodings generated by each model for the 8-digit code $86101521$.

| Encoder | Phrase |
|---|---|
| **Random** | meeting rawhide yelping hunch alum levy bog boom annum ivory gin sharing meme femme knock appeal sinning vivo readying bake twitch beaming pub hammock highlighting |
| **Unigram** | made right help enjoy william life back p.m. name over john sure mama foam neck able seen five right back touch p.m. baby make old |
| $n$-**gram** | Him to hire youth all. Be in show all my life. Be echoing be my own home. Of our age in which our. Him home of my own. Go up all his own. Of every day by god. Which by him by be. Him go along with. |
| **POS** | Matter with law by an age along mile of hope. Week by man among every age in age. Year among aim of man. Week by law use in favor with pike with age by humming by boy among week along the. |
| **Chunk** | Matter would leap new jail. Home life book by money home. Average enjoy her home movie. Human ego being less won five. Earth by god she poem by. Buy make lead. |
| **Sentence** | Matter tell been shell among life. Pickup man moving or which nature. Mama of many couples in favor. Tobacco touch pump by my cold. |

Table 3: Encodings generated by each model for the first 50 digits of $\pi$.

### 4.1 Comparison of Encodings

The **random encoder** generates mnemonic encodings that use obscure words without any structure. As such, the random encoder does not produce memorable encodings.

The **unigram encoder** improves on the random encoder by favoring more common words, which tends to result in shorter encodings. However, as neither model considers part-of-speech information, the two baselines produce unrelated sequences of words, which are difficult to chunk and to remember.

The $n$-**gram encoder** generates encodings that tend to be long and unmemorable. The encoder often produces incoherent phrases of common words, such as "the of which by his own." The sentences produced by the other three non-baseline models tend to be more memorable.

For example, the **POS encoder** generates longer, more syntactically plausible sentences. The biggest drawback of this model is the relative lack of verbs in its generated sentences.

The **chunk encoder** produces encodings similar to those produced by the POS encoder but with a noun-verb-noun pattern that results in relatively short, simply structured sentences. However, the chunk encoder has a slow running time due to the relatively expensive process of parsing each possible noun phrase and verb phrase.

The **sentence encoder** produces words such that each word encodes many digits and tends to be distinctive. This means that fewer words and thus fewer sentences are needed to encode a given sequence of digits, making the mnemonic encodings generated by the sentence encoder easy to chunk and memorable. We hypothesize that the sentence encoder generates more memorable sentences. We tested that hypothesis through a user study.

## 5 User Study

We conducted a user study to test the memorability of the phrases generated by our models. The study was presented as a study of *password mem-*

*orability*, in which each password was an 8-digit number or its encoding from the $n$-gram encoder or the sentence encoder. We compared the $n$-gram model (which served as our baseline) to the sentence model (which we hypothesized best balances all aspects of memorability) for four factors:

**Short-Term Recall:** How well can participants remember the number or its encoding after five minutes?

**Long-Term Recall:** How well can participants remember the number or its encoding after one week?

**Long-Term Recognition:** How well can participants identify the number or its encoding from a list of options after one week?

**Subjective Comparison:** How easy does each number or its encoding seem to users? This comparison may give an indication of how likely users are to consider trying a particular mnemonic.

### 5.1 Study Overview

The study was comprised of two online surveys, which together take about fifteen minutes to complete. We recruited participants through emails to a computer science summer research program and through social media posts. Participants were compensated with a five-dollar Amazon.com gift card via email after completing the second survey.

While we had 167 participants complete both surveys, we found in the second survey that 101 of the respondents were fraudulent. These responses shared a number of suspicious characteristics:

- They were completed consecutively, usually with only a few seconds from one response to the next, for both the first and the second survey.

- None of them spent as much time on the survey tasks as other participants. The median time spent on our distraction task, for example, was 4 seconds for these participants. For other participants, the median time spent on the same task was 81 seconds.

- They appeared in groups of consecutive email addresses from the same free email provider.

- They all remembered the numeric code and its encoding perfectly after one week.

Removing these participants left 67 responses to the first survey and 66 responses to the second.

### 5.2 Study Design

Each participant was randomly assigned to one of two groups, $n$-gram or sentence. Participants were identified by the email address they entered in the first survey, which we then used to send them a link to the second survey and to identify their responses.

The first survey presented each participant with an 8-digit sequence and a corresponding sequence of words from either the $n$-gram encoder or the sentence encoder that encoded the same 8-digit sequence. The participant was asked to remember both the number and the encoding, without writing them down, and was informed that they would be asked to recall both sequences at the end of the first survey and on the second survey. After entering both sequences to confirm initial memorization, the participant was asked to read a page of baking recipes for approximately five minutes. This page, containing numbers and words, served to clear the participant's working memory. Next, the participant was asked to enter both sequences, with an "I forgot" button available if necessary. Each correct or incorrect attempt was recorded, as was the time spent on each page of the survey.

The second survey was sent to each participant seven days after their completion of the first survey. The survey again asked each participant to recall both sequences and recorded the same information. The participant was then asked to recognize their 8-digit sequence from a list of five sequences and to recognize their word sequence from a list of five word sequences generated by their group's encoder. Next, the participant was shown an 8-digit sequence, its $n$-gram encoding, and its sentence encoding and asked to rank the three passwords from "easiest to remember" to "hardest to remember." Finally, the participant was asked to share the approach they used to memorize the two sequences.

### 5.3 Study Results

Our primary goal in performing this user study was to evaluate our hypothesis that the sentence encoder produced more memorable encodings than the $n$-gram encoder. We also sought

to determine whether the sentence encodings are more memorable than the relatively short 8-digit sequences themselves.

**Short-Term Recall:** On the first survey, more participants correctly confirmed the sentence password than the $n$-gram password. 29 of the 31 sentence password participants ($94\%$) remembered the password, as opposed to only 26 of the 36 $n$-gram password participants ($72\%$). This difference is statistically significant at $\alpha = 0.05$ under a z-test for two proportions with p = 0.02. Participants with the sentence password also spent significantly less time on the confirmation page than did participants with the $n$-gram password (27 seconds versus 52 seconds, with p = 0.019 under an independent two-tailed t-test). After reading the recipes distraction page, more participants recalled the numeric password than the $n$-gram password (55/67 versus 23/36, p = 0.0403).

**Long-Term Recall:** On the second survey, we found no statistically significant differences between the number of participants who recalled the $n$-gram password, the sentence password, or the numeric password (respectively 9/36, 10/30, 25/66).

**Long-Term Recognition:** On the second survey, more participants recognized the sentence password than the numeric password when shown five options for each (30/30 versus 58/66, $p = 0.05$).

**Subjective Comparison:** More participants rated the sentence password as "easiest to remember" than the numeric password (36/66 versus 24/66, $p = 0.04$), and more participants rated the numeric password as "easiest to remember" than the $n$-gram password (24/66 versus 6/66, $p < 0.001$).

The results of our user study indicate that the sentence encoder produces more memorable encodings than the $n$-gram encoder does. These results also indicate that $n$-gram encodings are harder to remember in the short-term than the number itself. While few participants recalled any passwords after seven days, more participants recognized the sentence password than the numeric password, indicating that the sentence password is more memorable. Furthermore, the sentence password was most frequently rated "easiest to remember," followed by the numeric password and the $n$-gram password. We conclude that the sentence encoder produces more memorable encod-

ings than the $n$-gram encoder and effectively aids in the memorization of numbers.

# 6 Conclusions and Future Work

We have described several systems for generating encodings of numbers using the major system. While $n$-gram models generate sentences that accurately match their training corpora, the sentences tend to be long and unmemorable. Sentences based on POS tags tend to be more memorable but are still not syntactically reasonable. Forcing the same sentence structure on every sentence by parsing ensures a reasonable structure but at a high computational cost. Ultimately, ensuring that each sentence is of a known but randomly selected syntactic structure that favors short encodings produces a reasonable balance of syntactic correctness, length, and memorability. A user study on password memorability supports our claim that the sentence encoder produces memorable mnemonic encodings of numbers.

Future work could further improve the sentence encoder. We could produce a more interesting variety of sentences by using punctuation from the training corpus besides periods, such as commas, exclamation marks, and question marks. The sentence encoder often produces a fragment as its last sentence because it runs out of digits to encode. This problem could be mitigated by making the encoder select shorter sentence templates when there are few digits remaining. Furthermore, the encoder could use more nuanced sentence templates to enforce subject-verb agreement and grammatical use of auxiliary verbs.

While the sentence encoder takes a greedy approach in an effort to encode digits in as few words as possible, another potential approach would be to use a dynamic programming algorithm to efficiently search through all possible encodings. Given a suitable objective function as a proxy for memorability, this could potentially produce a more memorable encoding than the sentence encoder without excessively increasing the encoding's length.

One issue in our user study was the unexpectedly short amount of time spent on the distraction page, with an average of 81 seconds instead of the intended five minutes. Future user studies should enforce a set distraction time through the use of a timer or other mechanism when studying short-term recall.

A future user study could examine the effectiveness of our system on longer numbers. While our user study did not show that any particular type of password was easiest to recall after seven days, we expect that a study involving longer passwords would show the sentence password to be easiest to recall. This is because the major system is well-suited for aiding the memorization of long numbers while the eight-digit numeric passwords used in our study are relatively short. It would also be interesting to see how memorable our encodings are in a context where users are prompted to recall the password each day over a long period of time.

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
