# Peer review of "Generating Memorable Mnemonic Encodings of Numbers"

_ACL 2017 — decision unknown_

[Official Review · Reviewer 1 · rating 2 · confidence 5]
soundness 5 · originality 3 · clarity 4 · impact 2 · substance 2 · appropriateness 3 · meaningful comparison 4 · presentation format Poster

This paper describes several ways to encode arbitrarily long sequences of
digits using something called the major system. In the major system, each digit
is mapped to one or more characters representing consonantal phonemes; the
possible mappings between digit and phoneme are predefined. The output of an
encoding is typically a sequence of words constrained such that digits in the
original sequence correspond to characters or digraphs in the output sequence
of words; vowels added surrounding the consonant phonemes to form words are
unconstrained. This paper describes several ways to encode your sequence of
digits such that the output sequence of words is more memorable, generally by
applying syntactic constraints and heuristics.

I found this application of natural language processing concepts somewhat
interesting, as I have not read an ACL paper on this topic before. However, I
found the paper and ideas presented here to have a rather old-school feel. With
much of the focus on n-gram models for generation, frequent POS-tag sequences,
and other heuristics, this paper really could have been written 15-20 years
ago. I am not sure that there is enough novelty in the ideas here to warrant
publication in ACL in 2017. There is no contribution to NLP itself, e.g. in
terms of modeling or search, and not a convincing contribution to the
application area which is just an instance of constrained generation. 

Since you start with one sequence and output another sequence with a very
straightforward monotonic mapping, it seems like a character-based
sequence-to-sequence encoder-decoder model (Sequence to Sequence Learning with
Neural Networks; Sutskever et al. 2014) would work rather well here, very
likely with very fluent output and fewer moving parts (e.g. trigram models and
POS tag and scoring heuristics and postprocessing with a bigram model). You can
use large amounts of training from an arbitrary genre and do not need to rely
on an already-tagged corpus like in this paper, or worry about a parser. This
would be a 2017 paper.

[Official Review · Reviewer 2 · rating 3 · confidence 4]
soundness 5 · originality 3 · clarity 5 · impact 3 · substance 3 · appropriateness 4 · meaningful comparison 4 · presentation format Poster

- Strengths:
This paper presents a sentence based approach to generating memorable mnemonics
for numbers. The evaluation study presented in the paper shows that the
sentence based approach indeed produces memorable mnemonics for short 8-digit
numbers (e.g. 86101521 --> Officiate Wasteland).
Overall the paper presents the problem, the background literature and the
solution in sufficient detail. Because memorizing numbers (e.g. phone numbers
and account numbers) is sufficiently common, this is an interesting problem.

- Weaknesses:
The proposed solution does not seem to scale-up well for longer numbers; seems
to work well with 8-digit numbers though. But many numbers that people need to
memorize such as phone numbers and credit card numbers are longer than
8-digits. Besides, a number may have a structure (e.g. a phone number has a
country code + area code + personal number) which people exploit while
memorizing numbers. As stated above, this paper addresses an important problem
but the current solution needs to be improved further (several ideas have been
listed by the authors in section 6).

- General Discussion:
The current presented approach, in comparison to existing approaches, is
promising.

[Official Review · Reviewer 3 · rating 3 · confidence 3]
soundness 5 · originality 3 · clarity 5 · impact 3 · substance 4 · appropriateness 5 · meaningful comparison 4 · presentation format Poster

- Strengths:

Tackles a not very explored task, with obvious practical application
Well written and motivated

- Weaknesses:

The only method of validation is a user study, which has several weaknesses.

- Discussion:

The paper investigates various methods to generate memorable mnemonic encodings
of numbers based on the “Major” system. As opposed to other methods that
rely on this system to encode sequences, the methods proposed in this work
return a single sequence (instead of a set of candidates) which is selected to
improve memorability. Since “memorability” is an ambiguous criterion to
optimize for, the authors explore various syntactic approaches that aim for
short and likely sentences.  Their final model uses a POS template sampled form
a set of “nice” structures, and a tri-gram language model to fill in the
slots of the template. 

The proposed approach is well motivated: the section on existing tools places
this approach in the context of previous work on security and memorability. The
authors point to results showing that passwords based on mnemonic phrases offer
the best of both worlds in terms of security (vs random passwords) and
memorability (vs naive passwords). This solid motivation will appease those
readers initially skeptical about the importance/feasibility of such
techniques. 

In terms of the proposed methods, the baselines and n-gram models
(unsurprisingly) generate bad encodings. The results in table 2 show that
indeed Chunk and Sentence produce shorter sentences, but for short digits such
as this one, how relevant are the additional characteristics of these methods
(eg. POS replacements, templates etc)? It seems that a simple n-gram model with
the number-of-digits-per-trigram reweighing could perform well here. 

The evaluation is weaker than the rest of the paper. My main concern is that a
one-time memorization setting seems inadequate to test this framework. Mnemonic
techniques are meant to aid recall after repeated memorization exercises, not
just a single “priming” event. Thus, a more informative setting would have
had the users be reminded of the number and encoding daily over a period of
time, and after a “buffer period”, test their recall. This would also more
closely resemble the real-life conditions in which such a technique would be
used (e.g. for password memorization).

In terms of the results, the difference between (long term) recall and
recognition is interesting. Do the authors have some explanation for why in the
former most methods performed similarly, but in the latter “Sentence”
performs better? Could it be that the use of not very likely words (e.g.
"officiate", in the example provided) make the encodings hard to remember but
easy to spot? If this were the case, it would somewhat defeat the purpose of
the approach.

Also, it would be useful for the reader if the paper provided  (e.g. in an
appendix) some examples of the digits/encodings that the users were presented
during the study, to get a better sense of the difficulty of recall and the
quality of the encodings. 

- Suggestions:

It would be nice to provide some background on the Major system for those not
familiar with it, which I suspect might be many in the ACL audience, myself
included. Where does it come from? What’s the logic behind those
digit-phoneme maps?